# Measurement and Simulation of Magnetic Properties of Nanocrystalline Alloys under High-Frequency Pulse Excitation

**DOI:** 10.3390/ma16072850

**Published:** 2023-04-03

**Authors:** Changgeng Zhang, Min Zhang, Yongjian Li

**Affiliations:** 1State Key Laboratory of Reliability and Intelligence of Electrical Equipment, Hebei University of Technology, Tianjin 300401, China; 2Province-Ministry Joint Key Laboratory of Electromagnetic Field and Electrical Apparatus Reliability, Hebei University of Technology, Tianjin 300401, China

**Keywords:** pulse excitation, nanocrystalline alloy, magnetic properties, finite element method (FEM), field-circuit coupling

## Abstract

In order to broaden the application of nanocrystalline soft magnetic materials in electrical engineering under extreme conditions, nanocrystalline alloys must also have good characteristics under high-frequency and nonsinusoidal excitation. In this paper, the magnetic properties of Fe-based nanocrystalline alloys excited by high repetition frequency pulses were measured. Excitation frequency and duty cycles are two important factors in the study of magnetic properties under pulse excitation. With the amplitude of the pulse remaining constant, different local hysteresis curves were obtained by changing the frequency and duty cycle. The experimental results proved that the higher the frequency is and the smaller the duty cycle is, the narrower the local hysteresis loop is. Finally, the finite element method (FEM) was used to model the magnetic core coupling with an impulse circuit based on the measured magnetic properties. Compared with the experimental results, the simulation results showed that the field-circuit coupling analysis model can effectively reflect the influence law of the frequency and duty cycle on magnetic properties.

## 1. Introduction

With the development of pulse power technology in civil and industrial applications, the requirements arouse intense interest in pulse magnetic components [1,2]. Especially in industrial applications, high output power and high pulse energy are not only the most desirable goals, but more important is the output pulse waveform and its repetition frequency. A pulse waveform directly affects whether the specific application can achieve the desired effect [3], and pulse repetition frequency usually determines the working rate and efficiency of the pulse equipment system [4]. High repetition frequency pulse power technology has a very broad application prospect in industry applications [5,6]. Recently, due to their high repetition rates, good stability, and long life, the pulse transformers [7,8] and magnetic switches [9] have been widely used in repetitive pulse power apparatuses for water treatment, the decomposition of harmful gases, food processing, electrostatic precipitators, etc. Magnetic core inductors and pulse current transformers are also widely used in pulse sources for pulse shaping and pulse monitoring, respectively [10,11]. However, from the current application of pulse power technology in the field of high and new technology and civil industry, there are few applications with commercial value, and many repetition frequency pulse power technology applications still remain in the experimental research and concept. The main reason is the lack of accurate measurements and simulations of the magnetic properties of the magnetic core materials under pulse excitation. The traditional magnetic properties of the magnetic materials are modeled by the data provided by material manufacturers excited with sinusoidal waves. As a result, it is bound to induce a large error in the simulation of the pulse power apparatus due to the insufficiency of the traditional magnetic properties. Therefore, it is necessary to test these magnetic core materials under pulse excitations in order to determine their characteristics in these specific applications.

At present, the measurement of the magnetic properties of ferromagnetic materials under pulse excitation mainly considers the actual working conditions of pulsed magnetic components, and the magnetic properties of ferromagnetic materials are measured under approximate operating conditions. Randy D. Curry et al. used the pulse-charged transmission line to design a nanosecond-level pulse power source and measured the magnetic properties of ferrite CN20 and FINEMET FT-1HS cores under an approximate square wave pulse excitation [12,13]. Yi Liu et al. designed a pulse source based on a magnetic pulse compression system, and they measured the magnetic properties of Fe-based nanocrystalline cores with different remanences excited by a cosine pulse [14,15,16,17]. In addition, through loading a periodic pulse current, Wei Xin et al. obtained the measurement results of the local hysteresis curve of No. 20 steel material under periodic pulse excitation and simulated the local hysteresis curves based on the Preisach hysteresis model [18,19]. Jaegu Choi et al. fabricated a low-inductance circuit to obtain the magnetic properties of a FINEMET FT-2H core under pulse excitation, and the Electromagnetic Transient Program (EMTP) simulation was carried out in order to determine the energy transfer in a three-staged magnetic pulse compressor system [20,21].

The magnetic elements in pulsed power equipment are usually made of soft magnetic materials. The soft magnetic material sample used in this study was Fe-based nanocrystalline (FE-N). As a new kind of soft magnetic material, nanocrystalline alloys have outstanding high-frequency magnetic characteristics compared with traditional soft magnetic materials because their strip thickness is only tens of microns [22,23]. Due to the high saturation of the magnetic flux density, low energy losses, low magnetostriction, wide adjustable range of permeability, and good temperature stability, the nanocrystalline alloy is often chosen as cores for pulse power apparatuses such as pulse transformers, magnetic switches, and magnetic core inductors [24]. It is promising that nanocrystalline cores employed in pulsed magnetic components can improve the energy transfer efficiency, reliability, and feasibility of pulse power systems.

In this paper, the dynamic magnetic properties of the FE-N under high-frequency pulse excitation were investigated theoretically and experimentally. The research presented will describe the construction of the test stand employed for data collection and the data analysis techniques used to generate the *B*-*H* curves. Additionally, the finite element method was used to establish the magnetic field and circuit analysis model of the FE-N magnetic core so as to calculate the magnetic flux density variation and spatial distribution of the magnetic core.

## 2. Measurement System

### 2.1. The Test Setup and the Measuring Principle

The toroidal sample used in this study was provided by the company Renqiu Yuda Electric Technology Co. (Cangzhou, China), which specializes in the production of amorphous and nanocrystalline materials. The sample was made of an Fe-based nanocrystalline, whose composition was Fe_73.5_CuNd_3_Si_13.5_B_9_. The outer radius *r*_o_ was 28.1 mm, the inner radius *r*_i_ was 22.6 mm, and the height *h* was 17.8 mm. The lamination factor *h* of the core was 0.8. The magnetic properties of the sample under 1 kHz sinusoidal excitation are described below. The saturation flux density *B*_s_ was 1.1 T. The relative permeability *μ*_r_ was about 30,000. The ratio between remanence *B*_r_ and *B*_s_ was 67%. The coercivity *H*_c_ was 5.1 A/m.

Figure 1 shows the schematic diagram of the test system. The excitation waveform was generated by the signal generator (Tektronix AFG 2021-SG, which was purchased from Corey Instrument, Shenzhen, China). The power of the signal generator was too low to fully magnetize the sample. The wideband power amplifier (ATA-4014, which was purchased from Aigtek, Xi’an, China) was used to amplify the pulse waveform and excite the toroidal sample. The current sensor (PEARSON Current Monitor, which was purchased from Shanghai Linhorn Electromechanical Technology Co., Shanghai, China) and the voltage divider (PINTECH DP-25, which was purchased from Guangzhou Deken Electronics Co., Guangzhou, China) measured the excitation current and induced voltage, respectively. The oscilloscope (KEYSIGHT InfiniiVision DSOX2024A, which was purchased from Shenzhen Kezhongke Instrument Co., Shenzhen, China) was used to detect the excitation waveforms and collect the excitation current and induced voltage at the same time.

According to Ampère’s circuital law [25], the applied magnetic field strength *H* in the core is calculated by
(1)$$H=\frac{N_{1}i\ln\left(r_{o}/r_{i}\right)}{2\pi \left(r_{o}-r_{i}\right)}=\frac{N_{1}i}{l_{m}},$$
where *N*_1_ is the number of turns in the exciting winding, *i* is the current flowing in the exciting winding, and *l*_m_ is the average magnetic circuit length of the toroidal sample. The magnetic flux density *B* in the core was mainly determined by Faraday’s law of electromagnetic induction [25]. When the effective cross-sectional area of the toroidal sample is *S*, the induced voltage in the measuring winding is
(2)$$u=\frac{d\psi }{dt}=N_{2}\eta h\left(r_{o}-r_{i}\right)\frac{dB}{dt}=N_{2}S\frac{dB}{dt},$$
where *y* is the flux linkage and *N*_2_ is the number of turns in the measuring winding. In order to determine the magnetic flux density *B* in the toroidal sample, the induced voltage *u* in Equation (2) shall be integrated:(3)$$B\left(t\right)=\frac{1}{N_{2}S}\int_{0}^{t}u\left(t\right)dt,$$

After *u(t)* is measured, the magnetic flux density *B* can be calculated.

### 2.2. Experimental Results and Analysis

Figure 2 shows the hysteresis loops when the sample is magnetized under sinusoidal excitation at 1 kHz. Figure 3a,b show the loss and magnetizing curves, respectively, when the sample is magnetized under sinusoidal excitation at different frequencies. As the frequency increases, the core loss increases and the permeability decreases, as shown in Figure 3. According to the Bertotti loss separation theory [26], the total loss *P*_tot_ is composed of the hysteresis loss *P*_hys_, eddy current loss *P*_eddy_, and anomalous loss *P*_anom_. With the increase in frequency, the *P*_hys_ remains unchanged while the *P*_eddy_ and *P*_anom_ increase, so the *P*_tot_ increases. In addition, the magnetic domains in the FE-N core cannot keep up with the rapid changes of the current in the excitation winding with increasing frequency, which results in a decrease in permeability.

Before the measurement of the magnetic properties under pulse excitation, the FE-N magnetic core is required to be demagnetized by the decaying sinusoidal excitation. The amplitude of the pulse voltage output by the power amplifier was 20 V. Figure 4 shows the local hysteresis loops when the magnetic core is excited by pulse excitation at different duty cycles. As can be seen from Figure 4, the shape of the local hysteresis loop is related to the frequency and duty cycle. When the duty cycle remains unchanged, the higher the frequency is, the narrower the local hysteresis loop is. In a similar way, the smaller the duty cycle is, the narrower the local hysteresis loop is. Because the amplitude of the applied pulse voltage remains constant, the oscillation of the magnetic flux density ∆*B* decreases with increasing frequency. Moreover, due to the permeability decreasing with increasing frequency, the blocking effect of the circuit on the excitation current increases, which leads to a corresponding decrease in the maximum magnetic field strength *H*_max_. When the frequency is constant and the duty ratio increases gradually, the integral value of the voltage *u*(*t*) in a cycle increases, so the ∆*B* increases correspondingly. In addition, the smaller the duty cycle is, the greater the effective frequency of the pulse excitation is, which results in a decrease in *H*_max_ accordingly.

## 3. The Simulation

The finite element simulation software COMSOL Multiphysics 5.3a, COMSOL AB, Stockholm, Sweden, was used to establish the magnetic field and circuit analysis model of the FE-N magnetic core. This paper demonstrates the coupling between transient simulations of the magnetic field and the pulse circuit. Including the effect of the hysteresis effects in the FE-N core, the model computes the spatial distribution of the magnetic flux density and the transient response of the core.

### 3.1. Modeling

The simulated experimental setup consisted of a toroidal core, which forms a closed magnetic flux path. The primary and secondary coils in the simulated experimental setup were wound uniformly around the core, as shown in Figure 5. Hysteresis effects are considered to simulate the magnetic behavior of the FE-N magnetic core. In the model, the primary and secondary windings are made of Litz wires, whose diameter is less than the skin depth. Thus, these windings are modeled as a muticoil conductor that neglects the eddy current effect. The primary winding is connected to a primary resistor, *R*_p_, and the pulse voltage source, *V*_p_, while the secondary winding is set to the open circuit, *I*_coil_ = 0 A. Several important design parameters such as the frequency and duty cycle of the input pulse are parameterized and therefore can be easily changed.

The Jiles–Atherton magnetic hysteresis model is used to simulate the characteristic of the magnetic core material, which has five unknown parameters: the saturation magnetization *M*_s_, the Langevin parameter *a*, the domain pinning parameter *k*, the reversible coefficient *c*, and the local field parameter *a*. The Jiles–Atherton model requires only a limited number of parameters: *a* and *M*_s_ to determine the slope of the hysteretic *B*-*H* loop at zero field and saturation, respectively; *c* and *k* are related to the strength of the hysteretic effects; and *a* is related to the remanence and d*B*/d*H* in the region near coercivity. Among the parameters, *M*_s_ can be directly measured or provided by its manufacturer. The other four parameters can be calculated iteratively from Equation (4) according to the magnetic susceptibility of the reference points on the static limit hysteresis loop and magnetization curve [27,28]. In this step, the reference points selected include the starting point of the magnetization curve, the remanence point (*M*_r_), the coercivity point (*H*_c_), and the loop tip (*H*_m_, *M*_m_). The values of the initial normal susceptibility, $\chi_{in}$; the differential susceptibility at the coercive point, $\chi_{H_{c}}$; the differential susceptibility at remanence, $\chi_{r}$; and the differential susceptibility at the loop tip, $\chi_{m}$ can be calculated based on these reference points:(4)$$\left\{\begin{array}{l} \chi_{in}=\frac{cM_{s}}{3a}\\ k=\frac{M_{an}(H_{c})}{1-c}\left[\alpha +\frac{1}{\frac{\chi_{H_{c}}}{1-c}-\frac{c}{1-c}\frac{dM_{an}\left(H_{c}\right)}{dH}}\right]\\ M_{r}=M_{an}\left(M_{r}\right)+\frac{k}{\frac{\alpha }{1-c}+\frac{1}{\chi_{r}-c\frac{dM_{an}\left(M_{r}\right)}{dH}}}\\ M_{m}=M_{an}(H_{m})-\frac{(1-c)k\chi_{m}}{\alpha \chi_{m}+1} \end{array}\right.,$$
where *M*_an_ is the anhysteretic magnetization. In order to improve the calculation speed, the system of equations is eliminated first. Ostensibly, each of the four parameters can be expressed in terms of the other three parameters so that three equations can be eliminated and only one equation needs to be solved. However, the hyperbolic cotangent function of *M*_an_ requires two parameters, *a* and *a*, which makes the elimination of three parameters extremely difficult. Therefore, for the sake of feasibility, only two parameters are eliminated. Two equations are kept and solved by the given initial values. Considering that the FE-N core selected in this paper was a soft magnetic material, the domain pinning parameter *k* should be retained. Because the coercivity of the soft magnetic is close to *k*, it is convenient to assign an initial value to parameter *k* and verify the calculation results. The iterative process is shown in Figure 6. Step 1: input the calculated values obtained by the experiment. Step 2: attach initial values to variables *a* and *k*. The value of parameter *k* should be close to the coercivity. Step 3: Calculate *a* and *k* and judge whether they are both greater than 0 and whether *k* is close to the coercive force *H*_c_. If the above conditions are not met, return to step 2 and reattach the initial value until it is met. Step 4: Substitute the values of *a* and *k* to the system of equations to calculate the parameters *a* and *c*. Check whether *a* and *c* are both greater than 0. If not, return to step 2 until both are greater than 0. Step 5: output the values of parameters *a*, *k*, *a*, and *c*.

The values of the parameters for the Jiles–Atherton model are presented in Table 1.

So far, the establishment of the Jiles–Atherton model is realized.

### 3.2. Results and Discussions

In order to verify the accuracy of the model, the static limit hysteresis loop of the core is calculated. The comparison results of the simulated and measured values of the hysteresis loop model are shown in Figure 7. As can be seen from the comparison results of the hysteresis loop, the simulated values are in good agreement with the experimental measurements.

Figure 8 describes the induced voltage in the secondary winding and the currents flowing through the primary winding when the pulse frequency was 1 kHz and the duty cycle was 5%. It can be seen that the simulated induced voltage and excitation current waveforms were basically consistent with the measured waveforms. The rise and fall times of the pulse source in the simulation were set as 200 ns, which was also the rise and fall times of the pulse signal given by the signal generator. The pulse waveform in the experiment was not an ideal square wave, and its rise and fall times were difficult to define. Additionally, due to the limitation of the switch inside the power amplifier in the test system, the rise time and fall time of the pulse excitation in the experiment were bound to be longer than those in the simulation, which led to the rise and fall parts of the simulated excitation current waveform being steeper than the measured waveform. In addition, the parameters of the toroidal core in the simulation were set to be isotropic. However, in practice, the parameters of the nanocrystalline core were anisotropic due to the influence of processing technology. So far, there are no studies on measuring the anisotropic magnetic parameters of the nanocrystalline toroidal sample. As can be seen from Figure 7, the nanocrystalline core in the simulation could not be more finely modeled, which is also the reason why the simulation excitation current waveforms were different from the experimental waveforms. Last but not least, the induced voltage is changed from a positive to negative value when the excitation current began to drop. The excitation current waveform in the simulation decreased faster than that in the measurement, so the minimum induced voltage in the simulation was smaller than that in the measurement.

After several pulses, the magnetic core reached a state of saturation. Figure 9 shows the distribution of the magnetic flux density in the magnetic core at time *t* = 1.05 ms (left) and *t* = 5.05 ms (right) when the magnetic flux density reached the maximum value in the cycle. When the pulse frequency was 1 kHz and the duty cycle was 5%, the core reached saturation at the end of the fifth pulse excitation. The simulation also showed that the number of pulses required for the core to reach a saturable state varied with the change in the pulse frequency and duty cycle. When the duty cycle remained constant, the higher the frequency was, the more pulses were needed. Similarly, the smaller the duty cycle was, the more pulses were needed. For example, only three pulses were needed when the frequency was 500 Hz and the duty cycle was 5%. However, nine pulses were needed when the frequency was 1 kHz and the duty cycle was 1%.

The hysteretic behavior can be displayed by plotting the magnetic flux density as a function of the magnetic field strength during one cycle. Figure 10 shows the local hysteresis loop obtained by averaging the quantities on a cross-section of the core. It can be seen from the results that the simulated local hysteresis loop approximated the measured local hysteresis loop. As can be seen from Figure 8, it took longer for the simulated excitation current to drop to 0. Therefore, compared with the measured hysteresis loop, the descending branch of the simulated local hysteresis loop changed more gently, and the simulated local hysteresis loop had a larger knee area.

## 4. Conclusions

An experimental system of generating a pulse waveform was designed to measure the local hysteresis loops in various conditions. The measurement results showed that the shape of the local hysteresis loop was related to the frequency and duty cycle. The higher the frequency was and the smaller the duty cycle was, the narrower the local hysteresis loop was. Based on the magnetic field and circuit analysis model, the spatial distribution of the magnetic flux density and the transient response of the core was calculated. The simulated results showed that the number of pulses required to stabilize the local hysteresis curve varied with the change in the frequency and duty cycle. The higher the frequency was and the smaller the duty cycle was, the higher the number of pulses required. Meanwhile, the simulated induced voltage and exciting current waveforms were in good agreement with the measured waveforms. Moreover, the simulated local hysteresis loop was similar to the measured local hysteresis loop. In conclusion, the model presented in this paper can be used to simulate nanocrystalline alloys in pulse magnetic elements.

## Figures and Tables

**Figure 1 materials-16-02850-f001:**
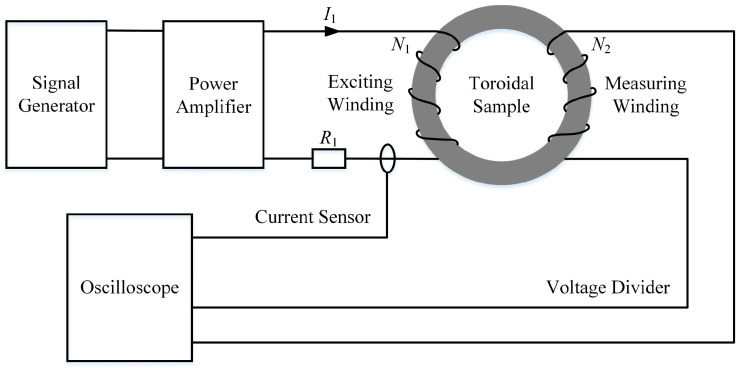
The configurations of the test system.

**Figure 2 materials-16-02850-f002:**
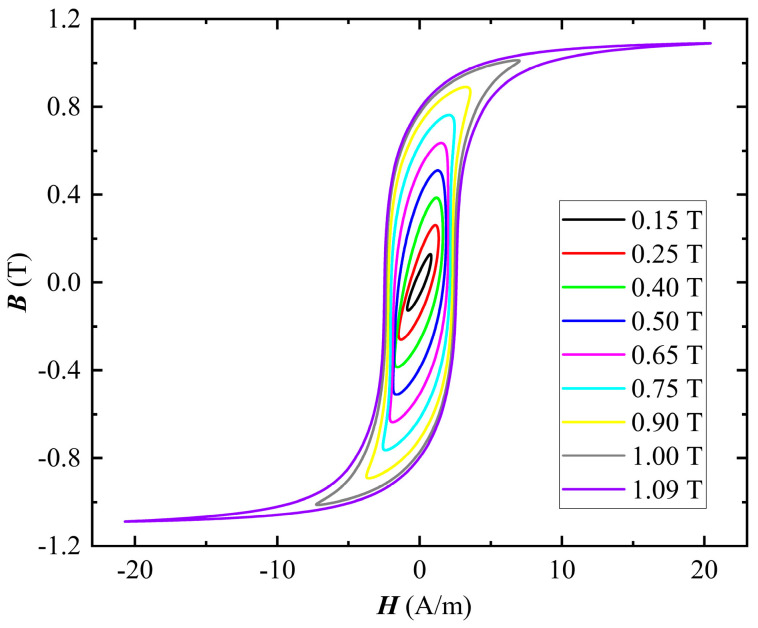
Hysteresis loops under sinusoidal excitation at 1 kHz.

**Figure 3 materials-16-02850-f003:**
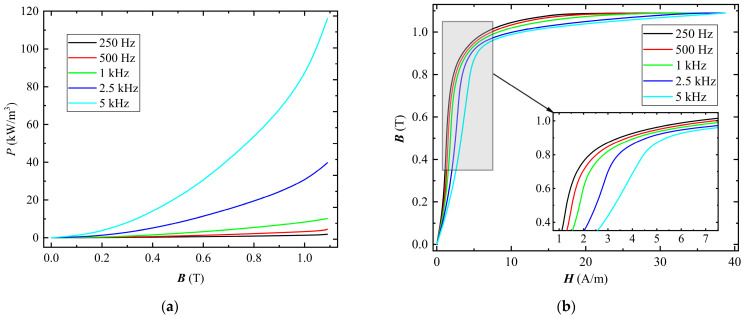
Magnetic properties of FE-N magnetic core under sinusoidal excitation at different frequencies: (**a**) core loss; (**b**) magnetizing curves.

**Figure 4 materials-16-02850-f004:**
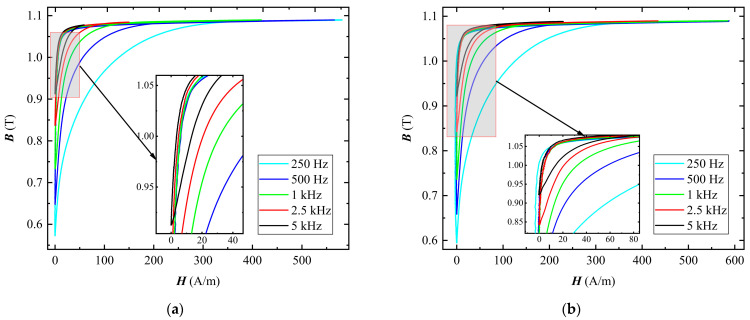
Local hysteresis loops under pulse excitation at different duty cycles: (**a**) *D* = 1%; (**b**) *D* = 5%.

**Figure 5 materials-16-02850-f005:**
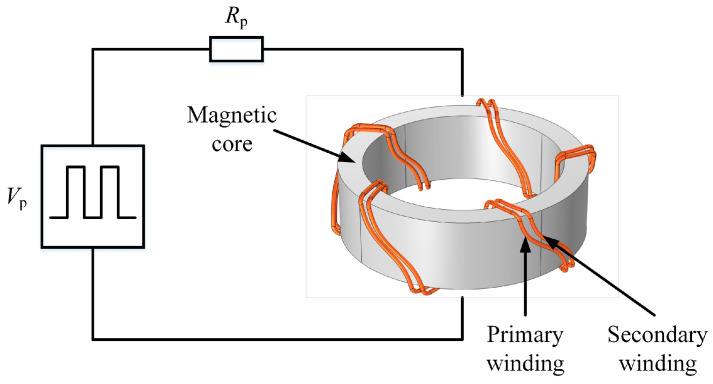
A magnetic core is connected to an external circuit with a voltage source and resistor.

**Figure 6 materials-16-02850-f006:**
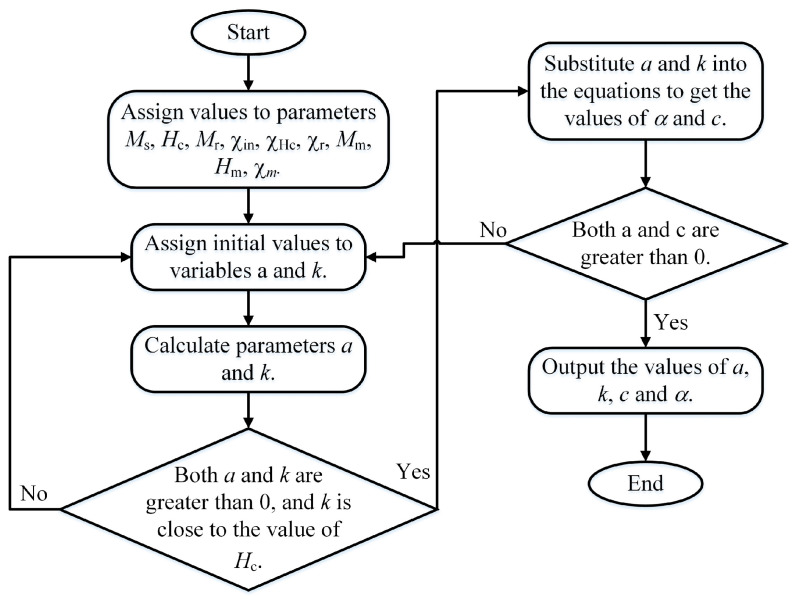
The process of solving basic parameters of the J-A hysteresis model.

**Figure 7 materials-16-02850-f007:**
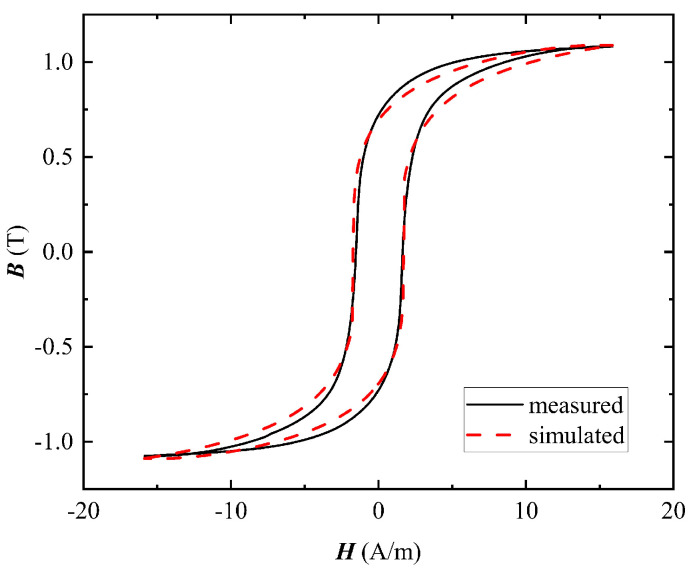
Comparison between the measured and simulated static limit hysteresis loop.

**Figure 8 materials-16-02850-f008:**
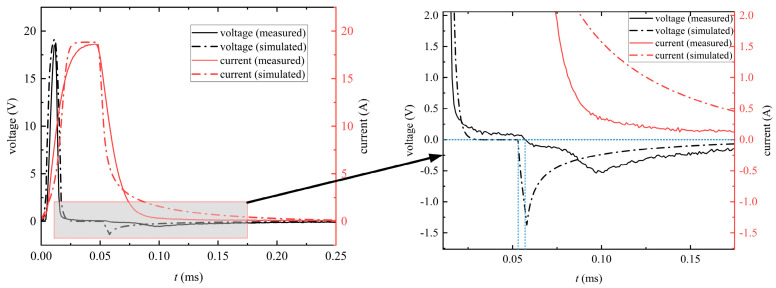
Comparison between the measured and simulated induced voltage and excitation current at *f* = 1 kHz and *D* = 5%.

**Figure 9 materials-16-02850-f009:**
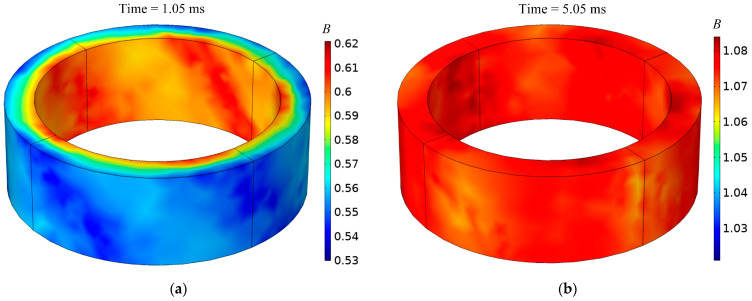
Magnetic flux density at different times: (**a**) *t* = 1.05 ms; (**b**) *t* = 5.05 ms.

**Figure 10 materials-16-02850-f010:**
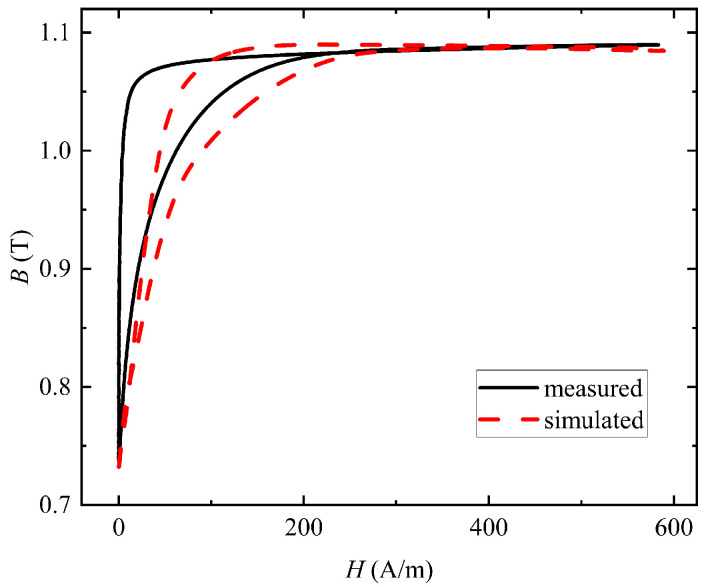
Local hysteresis loop in the core at *f* = 1 kHz and *D* = 5%.

**Table 1 materials-16-02850-t001:** Parameters for the Jiles–Atherton model.

Symbol Meaning	Symbol	Value
Saturation magnetization	*M* _s_	8.76 × 10^5^ A/m
Langevin parameter	*a*	4.94 A/m
Domain pinning parameter	*k*	4.99 A/m
Reversible coefficient	*c*	0.17
Local field parameters	*α*	2.37 × 10^−5^

## Data Availability

The data presented in this study are available on request from the corresponding author.

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
