# Peer review of "Measurement and Simulation of Magnetic Properties of Nanocrystalline Alloys under High-Frequency Pulse Excitation"

_materials, 2023, doi:10.3390/ma16072850_

Round 1
Reviewer 1 Report
I recommend to publish the manuscript after minor revision. Zhang and coworkers report on the experimental results and simulation of Fe-based nanocrystalline alloys excited by high repetition frequency pulses. Please see my comments below:
- It is not explained or described the way of obtention of the Fe-based toroidal sample employed and its compositional and magnetic characterization.
- In Line 93 before respectively it should be a comma.
- I suggest to add reference in line 115 for “Bertotti loss separation theory”.
- In line 168, as it refers to the 5 parameters, I suggest to modify as: “Among the parameters ...”
- It is not clear why the comparison between the experimental results and the simulation is only presented for the case of pulse frequency of 1 kHz and duty cycle of 5% and it also should be said on caption of Figure 8.
- In Figure 10 I suggest to add the experimental local hysteresis loop for better comparison as it is only said “approximates the measured local hysteresis loop”.
- The difference between the experiment and the simulated excitation current is not explained and also the way to correct it should be suggested. If could explain why the simulated local hysteresis loop approximates the measured while the current differs.
- If could explain the local minimum in the simulated results at approximately 0.06 ms in Figure 8.
Author Response
Dear Reviewer:
Thank you for your comments concerning our manuscript entitled “Measurement and Simulation of Magnetic Properties of Nanocrystalline Alloys under High-Frequency Pulse Excitation” (materials-2284907). These comments are all valuable and very helpful for revising and improving our paper, as well as the important guiding significance to our researchers. We have studied the comments carefully and have made corrections which we hope meet with approval. Please see the attachment for the main corrections in the paper and the responses to your comments.

Reviewer 2 Report
Upon initial evaluation, the manuscript is deemed not acceptable in its current form for publication in Materials journal. There are several areas that require significant revision and clarification. Firstly, it is unclear why the authors chose Fe73.5CuNd3Si13.5B9 alloy for this study and whether there are any prior experimental studies supporting their results. Additionally, the process of creating the toroidal sample used in the study and its key physical properties need to be described in detail. Secondly, it is unclear how the excitation waveform is generated, and what is the role of the wideband power amplifier in the test setup. Thirdly, there is a lack of information on the test setup and measuring principle used in the study, specifically on how the magnetic properties of the sample are measured under different conditions. Fourthly, the formula used to calculate the induced voltage in the measuring winding needs to be specified, along with its relationship to the magnetic flux density in the toroidal sample. Fifthly, the authors need to explain how the magnetic field strength in the core of the toroidal sample is calculated according to Ampere's circuital theorem. Lastly, it is necessary to provide detailed information on how the magnetic flux density in the toroidal sample is determined from the measured EMF. Therefore, the manuscript requires significant revisions before it can be considered for publication in Materials journal.
Author Response
Dear Reviewer:
Thank you for your comments concerning our manuscript entitled “Measurement and Simulation of Magnetic Properties of Nanocrystalline Alloys under High-Frequency Pulse Excitation” (materials-2284907). These comments are all valuable and very helpful for revising and improving our paper, as well as the important guiding significance to our researchers. We have studied the comments carefully and have made corrections which we hope meet with approval. Please see the attachment for the main corrections in this paper and the responses to your comments.
